# Parametric Mapping Cardiac Magnetic Resonance Imaging for the Diagnosis of Myocarditis in Children in the Era of COVID-19 and MIS-C

**DOI:** 10.3390/children9071061

**Published:** 2022-07-16

**Authors:** Bibhuti B. Das, Jyothsna Akam-Venkata, Mubeena Abdulkarim, Tarique Hussain

**Affiliations:** 1Department of Pediatrics, Children’s of Mississippi Heart Center, University of Mississippi Medical Center, Jackson, MS 39216, USA; jakamvenkata@umc.edu; 2Pediatric Cardiology, Nicklaus Children’s Hospital, Miami, FL 33155, USA; mubeena.abdulkarim@nickalushealth.org; 3Pediatric Cardiology, Children’s Health, UTSW Medical Center, Dallas, TX 75235, USA; mohammad.hussain@utsouthwestern.edu

**Keywords:** myocarditis, children, COVID-19, multisystem inflammatory syndrome in children, mRNA COVID-19 vaccine, cardiac MRI

## Abstract

Myocarditis comprises many clinical presentations ranging from asymptomatic to sudden cardiac death. The history, physical examination, cardiac biomarkers, inflammatory markers, and electrocardiogram are usually helpful in the initial assessment of suspected acute myocarditis. Echocardiography is the primary tool to detect ventricular wall motion abnormalities, pericardial effusion, valvular regurgitation, and impaired function. The advancement of cardiac magnetic resonance (CMR) imaging has been helpful in clinical practice for diagnosing myocarditis. A recent Scientific Statement by the American Heart Association suggested CMR as a confirmatory test to diagnose acute myocarditis in children. However, standard CMR parametric mapping parameters for diagnosing myocarditis are unavailable in pediatric patients for consistency and reliability in the interpretation. The present review highlights the unmet clinical needs for standard CMR parametric criteria for diagnosing acute and chronic myocarditis in children and differentiating dilated chronic myocarditis phenotype from idiopathic dilated cardiomyopathy. Of particular relevance to today’s practice, we also assess the potential and limitations of CMR to diagnose acute myocarditis in children exposed to severe acute respiratory syndrome coronavirus-2 infections. The latter section will discuss the multi-inflammatory syndrome in children (MIS-C) and mRNA coronavirus disease 19 vaccine-associated myocarditis.

## 1. Introduction

Myocarditis is an inflammatory disease of the myocardium and is a rare diagnosis in children accounting for only 0.05% of pediatric hospital admissions [1]. Acute myocarditis has a heterogeneous clinical course ranging from the asymptomatic presentation and gradual onset of heart failure to fulminant myocarditis presenting as cardiogenic shock and sudden death [2,3]. The complexity of these divergent presentations is further compounded by the continuously evolving diagnostic criteria, making the diagnosis exceptionally challenging [2]. Children with acute myocarditis often present with resting tachycardia, chest pain, respiratory distress, abdominal pain, and vomiting [3,4,5,6]. Chest pain is a typical symptom of acute myocarditis in adolescents [7]. Electrocardiography (ECG) abnormalities are present in approximately 90% of children with myocarditis, including ST-T wave, changes, low voltage QRS complexes, and atrioventricular conduction delays [3,5,6]. Elevations in cardiac biomarkers such as troponin and brain-type natriuretic peptides are common but not specific for acute myocarditis in children [3,7,8].

Echocardiography is the primary tool to determine the presence of left ventricular (LV) wall motion abnormalities, pericardial effusion, valvular abnormalities, and function assessment [9]. Endomyocardial biopsy (EMB) is the “gold standard” to confirm myocarditis, but it is an invasive procedure with suboptimal sensitivity. The advancement of cardiac magnetic resonance (CMR) imaging has been helpful in clinical practice for diagnosing myocarditis. Reflecting this evolution, a recent Scientific Statement by the American Heart Association (AHA) suggested CMR as the current gold standard for diagnosing myocarditis [10]. In addition to tissue characterization, CMR is useful for evaluating the ventricular function, myocardial wall thickness, chamber dilation, and pericardial effusion.

Clinically acute myocarditis implies a short time elapsed from the onset of symptoms (generally <1 month and up to 8 weeks) [11,12]. Chronic myocarditis (course of disease ≥ 3 months) is a type of inflammatory cardiomyopathy with either a dilated or non-dilated phenotype that is often characterized by regional or global wall motion abnormalities and impaired LV function [10]. In general, the prognosis of acute myocarditis is good, but up to 30% of cases may progress to develop a dilated cardiomyopathy phenotype [13]. The relative roles of viruses, host genomics, and environmental factors in disease progression and recovery are still unknown [14]. Consequently, treatment strategies are not well established. Confirming the diagnosis of acute myocarditis in children by CMR with tissue characterization using inverse recovery acquisition findings of myocardial inflammation is an essential milestone in the right direction and a paradigm shift towards a decreased need for EMB [10]. However, standard CMR parametric mapping parameters for diagnosing myocarditis are not established in pediatric patients for consistency and reliability in the interpretation. This review highlights the unmet clinical needs for CMR parametric criteria for diagnosing acute and chronic myocarditis in children and differentiating dilated chronic myocarditis phenotype from idiopathic dilated cardiomyopathy (DCM). Of particular relevance to today’s practice, we also assess the potential and limitations of CMR to diagnose acute myocarditis in children exposed to severe acute respiratory syndrome coronavirus 2 (SARS-CoV-2) infection, the multi-inflammatory syndrome in children (MIS-C), and mRNA coronavirus disease 19 (COVID-19) vaccine-associated myocarditis.

## 2. CMR Findings of Myocardial Inflammation and Pathological Correlations

Tissue characterization by CMR parametric mapping allows quantifying myocardial changes based on T1 and T2 images for fibrosis, water content, and extracellular volume fractions (ECV), a surrogate of interstitial fibrosis (Figure 1A–C and Figure 2A–C). It enables differentiation between normal myocardial muscle, edema, and fibrosis, thereby serving as a virtual biopsy [14,15]. CMR allows evaluation of all myocardial segments, contributing to its superior sensitivity to EMB to detect myocarditis because myocardial inflammation often has a heterogeneous, “patchy” distribution. Late gadolinium enhancement (LGE) helps detect scarring and necrosis.

The diagnostic CMR criteria are derived from the common consensus among experts (Lake Louise Consensus (LLC) criteria) for myocarditis in adults [16]. The hallmark features of myocardial inflammation on CMR are (1) T2-weighted imaging, which assesses for edema; (2) T1-weighted early gadolinium enhancement imaging (EGE), which assesses for hyperemia; and (3) LGE, which assesses for myocyte necrosis and scar. The CMR LLC criteria seem to have moderate accuracy in diagnosing acute myocarditis [17,18]. LLC criteria are qualitative analyses and rely upon the increased signal intensity of the myocardium relative to the normal myocardium. These criteria, therefore, have limited applicability for diffuse myocardial involvement without focal findings. Subsequently, a revised version of the LLC was released in 2018 with quantitative tissue mapping techniques. According to the updated LLC criteria, myocarditis can be diagnosed with high specificity based on: (i) the presence of one of the T2-based criteria (i.e., regional or global increase in T2 relaxation time or increased signal intensity on T2 weighted images) and (ii) the presence of one T1-based criterion out of three: prolonged T1 relaxation time, elevated ECV or LGE [19]. Both criteria have a reported median area under the curve for detecting myocarditis ranging from 0.75 to 0.90. Either isolated T1- or T2-based criteria in the appropriate clinical setting can suggest myocardial inflammation in adults [19,20]. It is important to note that the LLC criteria have not been validated in children.

The parametric mapping techniques have only recently begun to enter clinical use in children, and their predictive utility in acute myocarditis remains an active area of investigation. There is no comprehensive data for CMR based on the revised LLC criteria in pediatric myocarditis except for small observational studies. A study on 43 consecutive clinically suspected pediatric acute myocarditis patients demonstrated that the revised LLC criteria enhance the diagnostic performance compared to conventional LLC criteria [21]. In another retrospective study in 58 children with acute clinical myocarditis, LGE was identified as a predictor of poor outcomes [22]. Conventional LGE imaging can underestimate the amount of myocardial injury in myocarditis. Native T1 and ECV maps may reveal hidden myocardial injury in the normal-appearing myocardium of patients with myocarditis.

In one study, CMR was reported to have a low yield in diagnosing the severity of acute myocarditis and does not predict outcomes in children [23]. However, this study was likely underpowered to detect the outcome. Due to variability between individual scanners and local site references, there are significant differences in CMR tissue characterization techniques among pediatric centers and no standardized cutoff values for T1 or T2 relaxation times. Most centers use LGE for diagnosing scar or myocardial necrosis [24]. The presence of LGE is likely a robust prognostic marker in children and adults with myocarditis [25,26].

Subepicardial inferolateral LGE is the most common pattern in adult patients with acute myocarditis [27]. In contrast, a higher prevalence of a mid-wall or mixed LGE pattern is reported in pediatric patients [27]. The mid-wall or mixed pattern of LGE was associated with more severe disease progression and a higher complication rate than the subepicardial inferolateral pattern [28,29]. The LGE distribution pattern may help differentiate myocarditis from ischemic myocardial injury as seen in the anomalous left coronary artery from the pulmonary artery (ALCAPA), congenital coronary ostial stenosis or atresia, coronary fistula, or Kawasaki disease (KD) [30,31]. LGE involves the subendocardium following a coronary territory in cases of ALCAPA, and KD confers to an infarction-like pattern. In more severe cases, LGE may extend to transmural, resulting in a fully transmural pattern (Figure 3).

LGE patterns for viral myocarditis are often confined to the intramural septal wall or subepicardial lateral wall (Figure 4A,B), but the presence of LGE alone does not inform the chronicity of the disease process. Furthermore, Mahrholdt et al. [32] described different LGE patterns and correlated them with viral etiologies of acute myocarditis in children and young adults. Parvovirus B19 myocarditis had LGE in the inferolateral segments. In contrast, human herpes simplex type 6 (HHV6) myocarditis had LGE in the intraventricular septum, and the difference in distribution was due to different viral cardiac tropisms. A summary of CMR findings in children with either clinically suspected or EMB proved acute and chronic myocarditis in children are described in Table 1.

Patients with DCM often have a linear mid-myocardial LGE pattern, but a subendocardial pattern strongly suggests previous myocardial ischemia or necrosis due to microthrombus but spontaneous coronary recanalization or distal embolization from minimally stenotic coronary lesions [33]. In contrast, chronic myocarditis rarely presents with a subendocardial LGE pattern and rarely follows coronary artery distribution. Instead, the hallmark LGE pattern of chronic myocarditis consists of patchy or longitudinal striae of mid-wall and subepicardial enhancement. In chronic myocarditis, parametric tissue mapping to evaluate myocardial T1 relaxation time and ECV helps detect diffuse fibrosis without focal LGE [34].

## 3. CMR in Acute Myocarditis

It has been more than 30 years since CMR diagnosis of acute myocarditis was described in children [35]. Since then, the CMR technology has advanced, and more evidence of its clinical utility has been reported and is now considered a confirmatory test for diagnosing myocarditis in children [10]. However, the diagnostic value of CMR employing the 2018 LLC criteria in pediatric and adolescent patients with acute myocarditis is not externally validated. Furthermore, Lurz et al. assessed the performance of CMR and compared it with EMB in 129 consecutive suspected acute myocarditis patients, reported that the vast majority of patients had an excellent prognosis [37]. Thus, diagnosing acute myocarditis in this group, while intellectually engaging, did not alter clinically as most patients were managed with supportive care. It is unknown whether treatments with anti-inflammatory agents (prednisone, interferon, etc.) or immunosuppressive agents (azathioprine, sirolimus) directed at myocardial inflammation detected on CMR imaging can improve early functionality after acute myocarditis.

Furthermore, CMR cannot differentiate three significant myocarditis types: lymphocytic, giant cell, and eosinophilic myocarditis, where specific therapy could improve myocarditis and myocardial function [38]. Eosinophilic myocarditis is often easily reversible with cessation of the offending allergen and early use of corticosteroids. The CMR findings of circumferential LV subendocardial enhancement provide supportive evidence of eosinophilic myocarditis in the presence of eosinophilia and allow for the prompt institution of measures to promote recovery and alter its prognosis [39]. Whether CMR imaging alone is sufficient to initiate active treatment or guide immunosuppressive therapy in lymphocytic myocarditis remains controversial. Frustaci et al. demonstrated significant improvement in myocardial function after prednisone and azathioprine immunosuppression in 85 pathogen-negative biopsy-proven lymphocytic myocarditis patients [40]. Baccouche et al. compared CMR and EMB in consecutive patients with acute myocarditis in adults and suggested that combining CMR and EMB yields a considerable diagnostic synergy by overcoming the limitations of CMR and EMB as individually applied techniques [41]. This hybrid approach reduces sampling error, thereby increasing the sensitivity of EMB and making more detailed histopathological and etiological information available for decision-making.

There is a bimodal distribution of acute myocarditis in the pediatric population, with peaks during infancy and adolescents [2]. Besides the patient’s age, other factors may influence the CMR parameters for T1 and T2 relaxation times, e.g., heart rate, respiratory movements, or hydration status. Several practical considerations related to CMR must be taken, especially for routine CMR for suspected acute myocarditis in infants and young children, as they may not comply with breath holding during image acquisition. Each center has strategies to minimize generalized and cardio-respiratory motion artifacts depending on the child’s age, clinical condition, and available expertise and resources. Young children often require deep sedation and anesthesia to prevent artifacts and poor-quality CMR imaging [42,43]. CMR under general anesthesia is resource-intensive, requiring a pediatric anesthesiologist with cardiac experience and CMR-compatible equipment [44,45]. Since children with acute myocarditis are often very sick and pose a high risk for anesthesia, continuing intensive monitoring despite limited access to the child during the scan poses an additional challenge. Nonetheless, CMR seems to have an excellent safety profile even in these children when managed by trained personnel, good communication, and a comprehensive emergency plan [46,47].

## 4. CMR in Chronic Myocarditis

The European Society of Cardiology (ESC) Working Group on myocardial and pericardial diseases described three inflammatory cardiomyopathies: viral myocarditis, autoimmune myocarditis, and viral-induced immune myocarditis [13]. Despite significant advancements in the CMR-based diagnosis of myocarditis, there remain challenges in differentiating between acute and chronic myocarditis and distinguishing the three crucial types of myocarditis. In patients with acute symptoms, parametric mapping techniques provide a valuable tool for confirming or rejecting the diagnosis of myocarditis [48,49]. However, only T2 mapping has acceptable diagnostic performance in patients with chronic myocarditis [37]. The role of finding LGE by CMR in chronic myocarditis is controversial, and the implication of LGE findings in children differs from adults. In children and adolescents with DCM, LGE is rarely detected by CMR despite marked LV dilatation and severely depressed LV function, which undermines the role of LGE [50]. In contrast to DCM, in a study of 222 consecutive adults with biopsy-proven myocarditis, the presence of LGE was found to be the best independent predictor of all-cause mortality and cardiac mortality [51]. But in another study, CMR evidence of LGE in the acute phase of myocarditis was not a predictive marker of recovery in children with follow-up CMR [52]. LGE could completely disappear during the healing phase of acute myocarditis in children. In addition, LGE in the mid-mural location is often found in DCM unrelated to inflammation and is associated with ventricular tachycardia, increased morbidity mortality, and decreased response to HF therapy in adults. Conversely, CMR imaging and texture analysis of myocardial T1 and T2 maps can differentiate acute and myocarditis and is superior to LLC or averaged myocardial T1 and T2 values [53]. 

There is no agreement on the time interval to perform follow-up CMR or the duration of follow-up after an episode of acute myocarditis in children. It is essential to differentiate between inflammatory cardiomyopathy (subacute/chronic myocarditis with abnormal ventricular function) from idiopathic DCM as there is a significant difference in prognosis and outcomes [54]. Figure 5A–C and Figure 6A–C describe the T1 and T2 mapping images in a 12-year-old child with a history of Parvovirus B19 infection who presented with dilated LV with markedly decreased function and symptomatic heart failure. T1 mapping indicated a global increase in T1 relaxation time, but as per revised LLC criteria, the sensitivity of CMR drops as edema is less prominent than in acute myocarditis and even disappears entirely in most cases of chronic myocarditis [26,55]. In contrast to edema, LGE generally persists in most cases. However, its extent is slightly reduced from 6.2% to 4.1% of LV mass after six months, according to data from 187 patients in the Italian Multicenter Study on Acute Myocarditis Registry [56].

## 5. CMR in COVID-19-Associated Myocarditis

The challenges in diagnosing acute myocarditis were highlighted during the recent COVID-19 pandemic. In an extensive study of 718,365 COVID-19 patients from a global federated health research network, 5% had new onset myocarditis, and those who presented with myocarditis had increased odds of major adverse cardiac events [57]. The incidence of myocardial injury was 19.7% among 416 hospitalized COVID-19 patients in Wuhan, China [58]. The true epidemiology of COVID-19-related myocarditis in children is unknown because it is difficult to prove that SARS CoV-2 is a cardiotropic virus. However, coronavirus is known to induce myocarditis, though not being among the most commonly involved viral agents. One case report showed localization of SARS-CoV-2 in myocardial interstitium in an adolescent who died from acute fulminant myocarditis and cardiogenic shock [59]. In an autopsy series in adults, Lindner et al. [60] found that 24 (50%) had SARS-CoV-2 viral particles in their myocardial tissue samples but showed no evidence of myocardial inflammation. However, Halushka et al. [61] reviewed 22 publications describing autopsy results in 277 adult patients who died of COVID-19 and concluded that myocarditis was infrequent (1.4%) in this population. EMB is not recommended to diagnose COVID-19 myocarditis because the findings of macrophage infiltration are not specific for COVID-19 myocarditis, and isolation of SARS-CoV-2 is rare [62].

The mechanism of myocardial injury due to COVID-19 is not well characterized. Possible direct myocardial involvement and myocarditis through binding of SARS-CoV-2 spike protein to angiotensin-converting enzyme 2 (ACE2) protein, followed by endocytosis of the virus and subsequent viral replication and ACE2 down-regulation. Lack of ACE2 may decrease the conversion of Angiotensin II (ATII) to angiotensin, leading to an accumulation of ATII and an increase in its harmful effects on the cardiovascular system. Activated T-cells are responsible for cell-mediated cytotoxicity. Another possible mechanism for cardiovascular morbidities in patients with COVID-19 is the systemic inflammatory response, which may trigger hypoxia, ischemia, and vasculitis involving coronary arteries [63,64,65,66]. Noteworthily, cytokine storm, known to exacerbate the clinical course of COVID-19, promotes the activation of T-cells, which releases cytokines to maintain the exaggerated immune response. A schematic diagram of the putative mechanism of SARS-CoV-2 viral-induced myocarditis is shown in Figure 7.

CMR findings in COVID-19-related acute myocarditis cases do not differ from what is described in LLC criteria [67]. The Society for Cardiovascular Magnetic Resonance Imaging has proposed a protocol to evaluate myocarditis in COVID-19 infection [68]. Out of 100 adults who recently recovered from COVID-19, Puntmann et al. [69] identified cardiac involvement in CMR in 78%, but the case selection of this cohort (with many patients having new or persistent symptoms) influenced the high incidence. Many case reports and meta-analyses [70,71,72,73,74,75] describing the CMR findings in adolescents and young adult athletes aged 16 to 21 years who had positive for COVID-19 were published and summarized in Table 2. An example of COVID-19 myocarditis in a 15-year-old soccer player with linear subepicardial LGE is shown in Figure 8. There are no uniform CMR criteria to consistently diagnose acute myocarditis in COVID-19 patients, and conflicting CMR findings are reported in adults with COVID-19-associated myocarditis. One study showed the presence of diffuse myocardial edema in symptomatic COVID-19-associated myocarditis patients [76], whereas, in another study, there was less myocardial edema in COVID-19-associated myocardial injury compared to all-cause myocarditis [77]. Future studies with standardized CMR protocols are needed to investigate the long-term cardiovascular consequences of COVID-19.

## 6. CMR in Myocarditis with MIS-C

The pathophysiology of MIS-C is not fully understood. Preliminary studies suggest that a cytokine storm with a vasculitis process, myocardial edema, and microvascular disease are likely to contribute [78,79]. The CMR is only indicated when there is cardiac dysfunction by echocardiography and does not improve with a decrease in inflammatory markers. Therefore, in most cases, the CMR was performed at variable times from the initial clinical presentation. Bartoszek et al. [80] reported normal CMR two months after COVID-19-related MIS-C in 19 children with initial LV dysfunction by echocardiography. Webster et al. [81] reported normal CMR in 20 children without evidence of initial LV dysfunction at three months follow up after COVID-19 MIS-C. A multicenter international study of 111 children diagnosed with MIS-C showed that 20 (18%) patients had CMR criteria for acute myocarditis (as defined by LLC), and LGE was present in 18 of those 20 patients [82]. The authors of the study suggested that CMR helps identify a subset of MIS-C patients at risk for cardiac sequelae [82]. Other studies have shown diffuse myocardial edema on T2 inversion recovery sequences and native T1 mapping with no evidence of LGE one week after onset of symptoms, suggesting that CMR parametric mapping helps diagnose acute myocarditis in MIS-C in children [83]. On the other hand, another study in children where CMR was performed on 21 MIS-C patients six months after discharge from the hospital showed no evidence of edema in T2-weighed sequences, and three patients had LGE [84]. Figure 9 describes CMR findings of elevated T1 relaxation time but no LGE in an 8-year-old girl three months after MIS-C. The difference in variability of CMR findings in MIS-C patients might be related to the timing when CMR was performed. Normal CMR was reported in children during mid-term recovery from MIS-C [81].

In a few studies, CMR showed no evidence of LGE, and there was no correlation between LGE and LV dysfunction. However, another study demonstrated a good correlation between CMR-based myocardial parametric mapping and echocardiography-based global longitudinal strain and strain rate [85]. The authors of this study also showed that none of the 70% of MIS-C cases with biochemical evidence of myocardial injury at presentation had residual LGE on follow-up at three months [85]. CMR can be helpful in both acute and convalescent periods, where echocardiography may provide inadequate images. Further studies are needed to determine the long-term functional implications of CMR findings in MIS-C.

## 7. CMR in mRNA COVID-19 Vaccine-Associated Myocarditis

Since April 2021, increased acute myocarditis cases have been reported in conjunction with the COVID-19 vaccinations, particularly among adolescents [86,87,88,89,90,91,92,93]. The pathogenesis of mRNA COVID-19 vaccine-associated transient myocarditis is unknown, but an antibody-mediated mechanism is speculated. However, Muthukumar et al. [94] explored the SARS-CoV-2 antibody levels in patients with acute myocarditis and failed to prove the hypothesis of antibody-mediated myocardial injury. The three main mechanisms by which COVID-19 mRNA vaccines might induce hyperimmunity are mRNA immune reactivity, antibodies to SARS-CoV-2 spike glycoproteins cross-reacting with myocardial contractile proteins, and hormonal differences [95]. Alternatively, the lipid nanoparticle sheath, a common structural component of m-RNA vaccine platforms, could be implicated in the pathogenesis of vaccine-induced myocarditis [96].

The published reports suggest myocardial Inflammation in many mRNA vaccine-associated myocarditis patients where CMR was abnormal as defined by the 2018 revised LLC criteria. The available studies indicate mild clinical symptoms and rapid resolution of myocarditis after mRNA vaccination. CMR showed LGE was common in these patients (94% had LGE), and normal LV systolic function was usual by an echocardiogram (92% had normal systolic function by echocardiogram) [88]. In this cohort, the pattern of LGE was largely subepicardial or mid-myocardial. In the short-term follow-up, all patients with vaccine-associated myocarditis reported in the cohort were asymptomatic with no adverse events. In another study of 69 patients with COVID-19 vaccine-associated myocarditis, most patients had normal LV systolic function with mild LV dysfunction noted in 14% (9 patients) [87]. In this study, most patients (88%) had myocardial edema and LGE. All of these patients with LGE had involvement in the subepicardial layer of the lateral and inferior wall of the LV [87]. The distribution of LGE in a 16-year-old male is shown in Figure 10. The long-term impact of LGE found by CMR in the setting of normal LV function by echocardiogram remains unknown and needs to be followed with repeat CMR in 6 to 12 months.

## 8. Conclusions

Polymerase chain reaction testing revealed a broad spectrum of viruses, including rhinovirus, influenza A/B, respiratory syncytial virus, coronavirus, adenovirus, coxsackie B5 virus, enterovirus, and parainfluenza virus in children with myocarditis in the pre-COVID-19 era [97]. In the last decade, the etiology of myocarditis has shifted dramatically, with parvovirus B19 and HHV6 replacing cases due to enterovirus and adenovirus. In the current era, the ongoing COVID-19 pandemic and the increasing number of myocarditis related to SARS-CoV-2 infection might bring about yet another etiological shift. Recently, CMR has emerged as an essential test because of its noninvasive nature, high sensitivity, and ability to comprehensively evaluate the myocardial function, structure, and tissue characterization. However, CMR findings and LGE extension can be a dynamic process and time-dependent in the acute phase of acute myocarditis. Per the consensus statement by the Society for Cardiovascular Magnetic Resonance [34], parametric CMR T1 and T2 imaging were superior to LGE in diagnosing and prognosis of acute myocarditis in adults. However, there is a need for external validation of CMR parametric parameters in children. The present review identified a crucial need for more in-depth information on CMR parametric imaging parameters to accurately diagnose and manage myocarditis in children and predict adverse long-term outcomes in patients with suspected acute myocarditis.

## Figures and Tables

**Figure 1 children-09-01061-f001:**
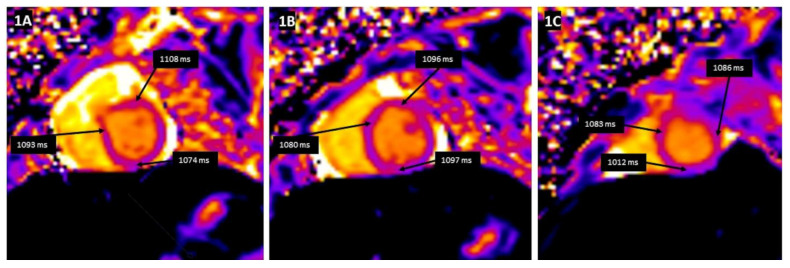
T1 mapping demonstrates a global increase in myocardial T1 relaxation times at the base (**A**), mid-ventricular level (**B**), and the apex (**C**). The average left ventricular myocardial T1 relaxation time is prolonged (1067 ms), and myocardial extracellular volume is elevated (32%).

**Figure 2 children-09-01061-f002:**
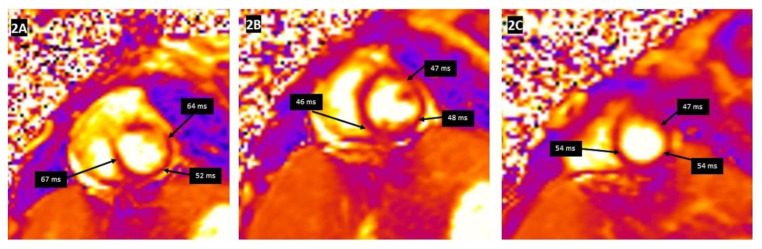
T2 mapping demonstrating the regional increase in myocardial T2 relaxation times at the basal anteroseptal and anterolateral segments (**A**), with normal T2 times at the mid-ventricular level (**B**) and the apex (**C**). Clinical vignette: A 20-month-old female with myocarditis associated with rhino/enterovirus, rapid left ventricular ejection fraction recovery from 24% to 48%, elevated troponin, and diffuse low voltage QRS complexes on electrocardiogram. There was no evidence of myocardial delayed enhancement after gadolinium contrast emphasizing the additional utility of T1 and T2 mapping in the evaluation of myocarditis.

**Figure 3 children-09-01061-f003:**
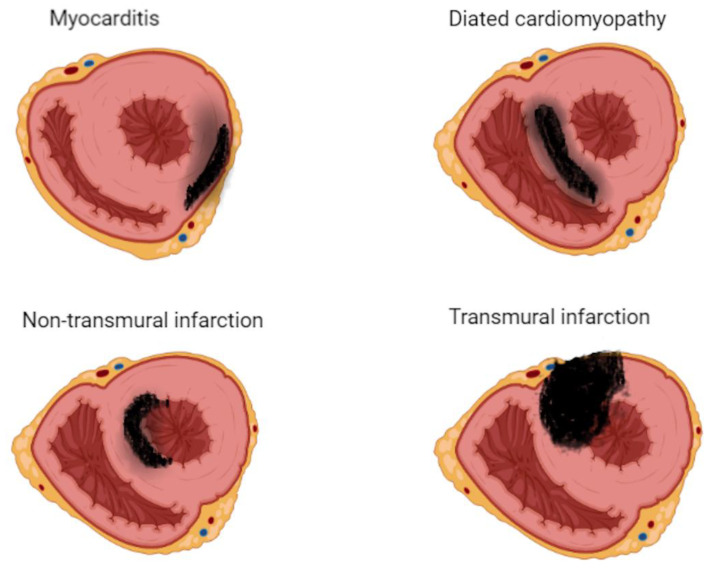
Late gadolinium enhancement (LGE) imaging allows the noninvasive visualization of areas affected by myocardial scar, conferring clinicians the unique ability to differentiate ischemic from nonischemic lesions based on typical LGE patterns.

**Figure 4 children-09-01061-f004:**
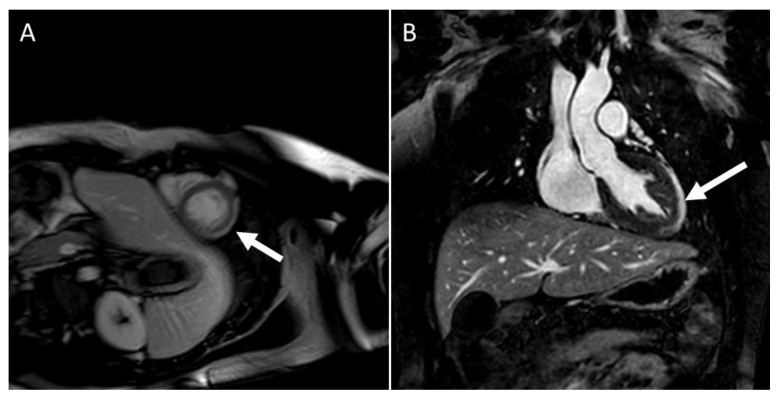
CMR in an adolescent with acute myocarditis: (**A**) Shows subepicardial early gadolinium enhancement (EGE) (white arrow); (**B**) Shows subepicardial LGE (white arrow).

**Figure 5 children-09-01061-f005:**
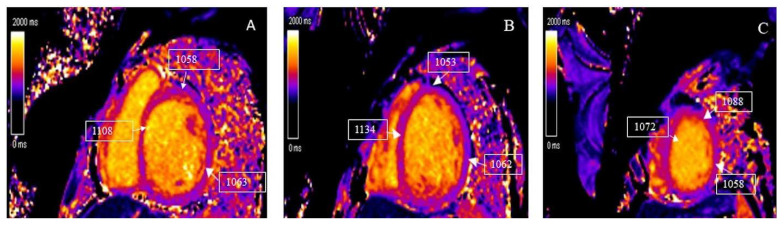
T1 mapping demonstrating the global increase in myocardial T1 relaxation times at the base (**A**), mid-ventricular level (**B**), and the apex (**C**). The average left ventricular myocardial T1 relaxation time is mildly prolonged (1077 ms).

**Figure 6 children-09-01061-f006:**
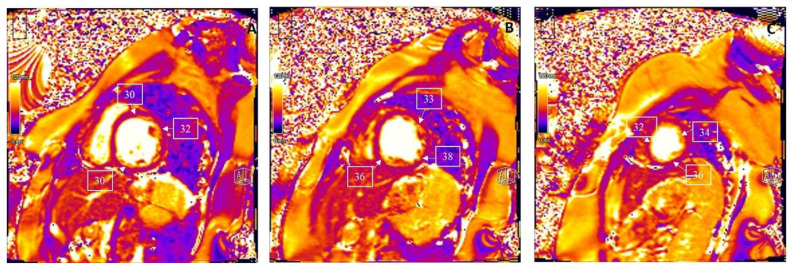
T2 mapping demonstrating normal myocardial T2 relaxation times at the base (**A**), mid-ventricular level (**B**), and at the apex (**C**). Clinical vignette: 12-year-old female with possible chronic myocarditis associated with Parvovirus B19, dilated LV with decreased LVEF 22%.

**Figure 7 children-09-01061-f007:**
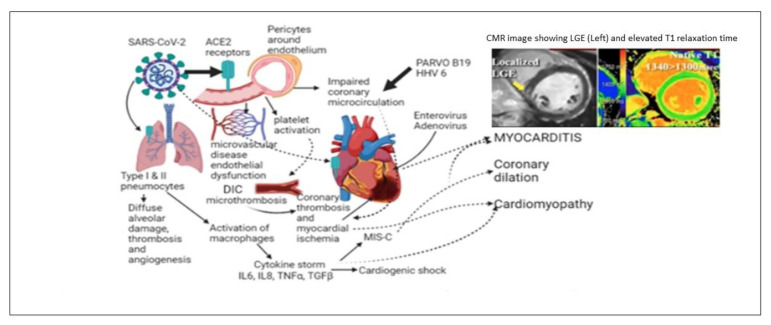
Several putative mechanisms of myocarditis due to SARS-CoV-2 and other common cardiotropic viruses.

**Figure 8 children-09-01061-f008:**
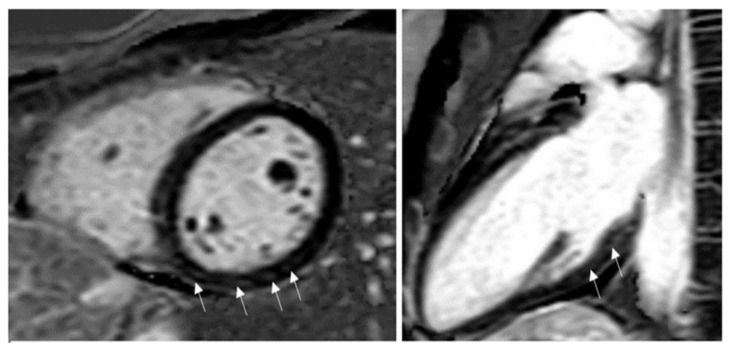
LGE in the mid myocardium of the inferior and lateral wall (white arrows). Clinical vignette: 15-year-old-female soccer player with COVID-19-associated myocarditis. This subject reported shortness of breath and chest pain with activity and was found to have ventricular ectopy and non-sustained runs of ventricular tachycardia on Holter monitoring.

**Figure 9 children-09-01061-f009:**
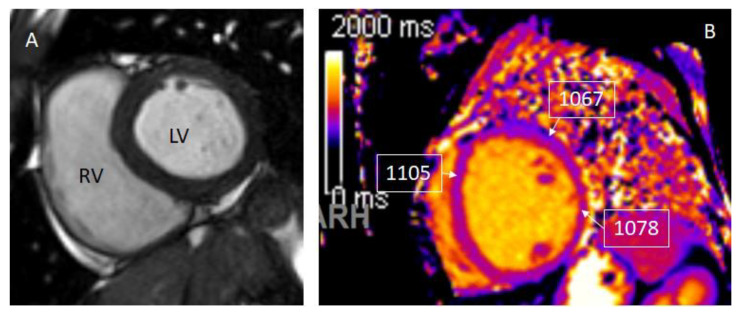
CMR shows no LGE (**A**) but diffusely elevated T1 time (**B**).

**Figure 10 children-09-01061-f010:**
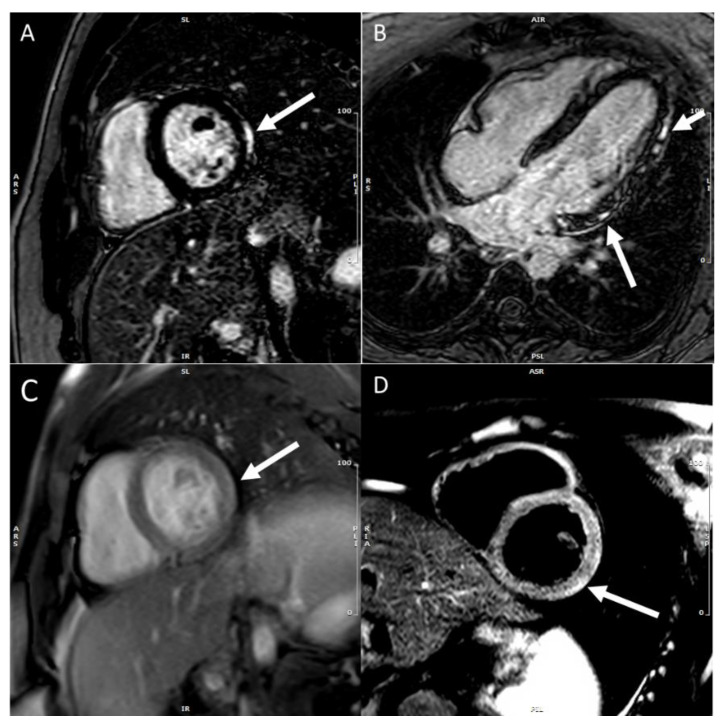
A 16-year-old-male with mRNA COVID-19 vaccine-associated myocarditis. (**A**) subepicardial LGE (white arrow); (**B**) subepicardial LGE (white arrow); (**C**) early gadolinium enhancement (EGE) (white arrow); and (**D**) shows enhanced T2 signal (white arrow).

**Table 1 children-09-01061-t001:** CMR findings in clinically suspected acute and chronic myocarditis in children.

Reference	*n*	Age	CMR in Days Following Acute Clinical Myocarditis	Abnormal T1 Plus T2 CMR Findings *n* (%)	LGE + *n* (%)
Gagliardi et al. [35]	11	9 mo– 9 yrs	24–48 days	Tissue characterization obtained in T1 weighed spin-echo sequences in 100% of pts	
Banka et al. [24]	143	16 yrs (mean)	2 days (mean)	LLC + ve in 117 (82%), negative in 18 (13%), And equivocal in 7 (5%) yielding a sensitivity of 82%	115 (81%)
Martinez-Villar et al. [7]	26	0–16 yrs (median) 14 yrs	11–53 days	Total of 2 of 3 LLC in all 26 patients (100%)	26 (100%)
Cornicelli et al. [22]	23	15.1 yrs (mean)	4.5 days (mean)	LLC: diagnostic yield 57% Revised LLC increased diagnostic yield to77%	86%
Chinali et al. [25]	40	2–17 yrs (median) 13 yrs	At admission 10/40 had FU CMR	Myocardial edema in 33 (82.5%) 6 recovered, 4(40%) had persistent fibrosis	19 (47.5%) 4(40%) had persistent LGE
Dubey et al. [36]	34 (Follow-up CMR in 12 who had LGE at baseline)	10–17 yrs (median 16 yrs)	After discharge		Persistence LGE in 10/12 (83%)
Martin et al. [28]	125	Average 15.1 yrs	Average 8 days	Revised LLC in 94 (76%) 79 had FU: 16 pts had disappearance of LLC	93 (74.4%) 35(28%) had persistent LGE
Isaak et al. [21]	43 (Follow-up CMR in 27/43 pts But only 17 had parametric mapping available)	8–21 yrs (mean) 17 yrs	1–9 days of initial diagnosis 53 days from the initial CMR (median)	Focal edema in 32 (74.4%) Persistent focal edema in 12/27 (44.4%)	36 (83.7%) LGE persistent in 20/27 (74%)

**Table 2 children-09-01061-t002:** CMR findings in children and young adults with COVID-19-associated myocarditis.

Reference	*n*	Age	CMR in Days after COVID-19 + Test	Clinical Symptoms	Abnormal T1 Plus T2 CMR Findings *n* (%)	LGE + *n* (%)
Gnecchi M, et al. [72]	1	16 yrs	2 days	Symptomatic	T2 mapping-patchy edema of the lateral wall	Subpericardial LGE+
Das BB [73]	1	16 yrs	60 days	Symptomatic	ECV 40% Relative myocardial signal intensity was calculated to be >2.4 compared to the pectoralis muscle	No LGE
Rajpal et al. [74]	26	19–21 yrs	11–53 days	46% symptomatic	4 (15%)	12 (46%)
Clark et al. [75]	59	19–21 yrs	13–37 days	78% symptomatic	16 (27%)	2 (3%)
Starekova et al. [76]	145	19–21 yrs	11–94 days	77% symptomatic	2 (1.4%)	42 (29%)
Kim In-C et al. [77]	1	21 yrs	NA	Symptomatic	Native T1 +ve for ECV 61%,	Transmural LGE +

## Data Availability

The figures used in this review paper are original and is available with the corresponding author.

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
