# Peer review of "Parametric Mapping Cardiac Magnetic Resonance Imaging for the Diagnosis of Myocarditis in Children in the Era of COVID-19 and MIS-C"

_children, 2022, doi:10.3390/children9071061_

Round 1

Reviewer 1 Report

I have read with interest the paper by Das and colleagues. In the very comprehensive review, they described CMR findings of myocardial inflammation, CMR features of both acute and chronic myocarditis, the role of CMR in COVID-19-associated myocarditis, myocarditis in MIS-C, and the role of CMR in COVID-19 vaccine associated myocarditis.

The paper is well written and presented. The quality of the figures is good. There isn’t much to criticize. I have only one minor remark:

-       Section 2: T2* images co not show either fibrosis, water content or ECV. Please correct.

Author Response

Thank you very much. We are thankful for your review and comments.

We corrected the mistake and removed T2* (which is different for iron content etc.)

Reviewer 2 Report

- Please find " Role of Cardiac Imaging Modalities in the Evaluation of COVID-19-Related Cardiomyopathy"

in Diagnostic, that I was able to review cause there are some interesting data about imaging modalities that could improve the introduction part of this MS. 

- "At present, some acute myocarditis patients recover without residual myocardial injury, whereas others develop a dilated phenotype." Provide data and cite. 

- "Almost all 129 patients diagnosed with acute myocarditis with CMR have recovered with supporting therapy" ¿completely?

- Please relate time of diagnosis and recovery

- Conclusion: A critical opinion about the possibilities of LGE should be done.

Author Response

"At present, some acute myocarditis patients recover without residual myocardial injury, whereas others develop a dilated phenotype." Provide data and cite. 

- Thank you for your review. We corrected the sentence and added citations as below.

Prognosis is generally good with recovery in most; however, up to 30% with biopsy-proven myocarditis progress to develop a dilated cardiomyopathy and its potential associated complications. 

"Caforio ALP, Pankuweit S, Arbustini E et al. Current state of
knowledge on aetiology, diagnosis, management, and therapy
of myocarditis: a position statement of the European Society of
Cardiology Working Group on Myocardial and Pericardial Diseases.
Eur Heart J 2013;34:2636–48" This reference is added to the revised manuscript.

"Almost all 129 patients diagnosed with acute myocarditis with CMR have recovered with supporting therapy" ¿completely?

  • Thank you for your comment: To clarify: Furthermore, a large study, where CMR and EMB are compared diagnosing suspected myocarditis in  129 consecutive patients, found that the vast majority of patients with acute onset myocarditis (which in this study is defined as <14 days) had an excellent prognosis.36 Thus, diagnosing acute myocarditis in this group, while intellectually interesting, did alter clinical management as most patients were managed with supportive care.

Please relate time of diagnosis and recovery

  • We have added time to recovery through the text whenever it is available. But in general, a lack of a standard time for follow-up is not available or not homogeneous as most studies are retrospective data analyses. Added this sentence in conclusion, "However, CMR findings and LGE extension can be a dynamic process and it is time-dependent in the acute phase of acute myocarditis.

Conclusion: A critical opinion about the possibilities of LGE should be done.

  • We added this sentence in the conclusion, "As per the consensus statement by the Society for Cardiovascular Magnetic Resonance,34 parametric CMR T1 and T2 imaging is superior to LGE in diagnosis and prognosis of acute myocarditis. "